# Precision Anti-Cancer Medicines by Oligonucleotide Therapeutics in Clinical Research Targeting Undruggable Proteins and Non-Coding RNAs

**DOI:** 10.3390/pharmaceutics14071453

**Published:** 2022-07-12

**Authors:** Damiano Bartolucci, Andrea Pession, Patrizia Hrelia, Roberto Tonelli

**Affiliations:** 1R&D Department, BIOGENERA SpA, 40064 Bologna, Italy; damiano.bartolucci@biogenera.com; 2Pediatric Unit, IRCCS, Azienda Ospedaliero-Universitaria di Bologna, 40138 Bologna, Italy; andrea.pession@unibo.it; 3Department of Pharmacy and Biotechnology, University of Bologna, 40126 Bologna, Italy; patrizia.hrelia@unibo.it

**Keywords:** cancer therapy, new anti-cancer drugs, precision medicine, undruggable targets, non-coding RNAs, oligonucleotide therapeutics

## Abstract

Cancer incidence and mortality continue to increase, while the conventional chemotherapeutic drugs confer limited efficacy and relevant toxic side effects. Novel strategies are urgently needed for more effective and safe therapeutics in oncology. However, a large number of proteins are considered undruggable by conventional drugs, such as the small molecules. Moreover, the mRNA itself retains oncological functions, and its targeting offers the double advantage of blocking the tumorigenic activities of the mRNA and the translation into protein. Finally, a large family of non-coding RNAs (ncRNAs) has recently emerged that are also dysregulated in cancer, but they could not be targeted by drugs directed against the proteins. In this context, this review describes how the oligonucleotide therapeutics targeting RNA or DNA sequences, are emerging as a new class of drugs, able to tackle the limitations described above. Numerous clinical trials are evaluating oligonucleotides for tumor treatment, and in the next few years some of them are expected to reach the market. We describe the oligonucleotide therapeutics targeting undruggable proteins (focusing on the most relevant, such as those originating from the MYC and RAS gene families), and for ncRNAs, in particular on those that are under clinical trial evaluation in oncology. We highlight the challenges and solutions for the clinical success of oligonucleotide therapeutics, with particular emphasis on the peculiar challenges that render it arduous to treat tumors, such as heterogeneity and the high mutation rate. In the review are presented these and other advantages offered by the oligonucleotide as an emerging class of biotherapeutics for a new era of precision anti-cancer medicine.

## 1. Introduction

Cancer is a disease of uncontrolled cell growth caused by various genetic (i.e., mutations, amplifications, deletions, and translocations) and epigenetic alterations (i.e., hypo- or hypermethylation), and characterized by dysregulation in multiple cellular signaling pathways involved in processes, including cellular proliferation, survival, cell death, differentiation, energy metabolism, genomic stability, DNA repair, and escape from immune surveillance [1]. Both intra-/inter-tumoral heterogeneities caused by genetic, epigenetic, and regional adaptive patterns are extremely high in some cancers. The molecular heterogeneity is considered as one of the major failures of cancer therapeutics, radiotherapy, and chemotherapy, and various responses range from no response due to intrinsic resistance to a complete response [2]. In addition, conventional untargeted chemotherapeutic strategies often increase the rate of cancer mutations, and impose a selection of tumor cells that become resistant to the chemotherapy, which could also create new mutations in the healthy cells, in addition to the toxicities and adverse side effects on healthy organs, such as bone marrow, kidneys, heart, brain, liver, eyes, and other normal tissues [3]. Cancer incidence and mortality continue to increase rapidly worldwide [4,5]. The conventional cancer chemotherapeutic drugs confer relevant toxic side effects [6,7,8], while their clinical efficacy is often obtained for limited periods of time, because drug resistance emerges and the tumors evolve [9,10]. Therefore, the development and discovery of novel therapeutic strategies are urgently needed to offer more effective and safe therapeutic options. Novel cancer therapy strategies emerged in the recent years with the potential to selectively detect and eradicate malignant cells, with minimal damage to the healthy tissue. In this context, many mutated genes that play a causative role in tumorigenesis have been identified and characterized in many cases, and could constitute the starting point for targeted and effective precision medicine therapies. However, it is estimated that a large number of proteins (about 70–85%) are considered undruggable, or difficult to be targeted by conventional drug discovery approaches, such as the small molecules [11,12,13]. These traditional approaches could not generate new drugs that enable a precision medicine that maximizes specificity against the target, while minimizing or eliminating the potential side effects and toxicity. The limitation of other relevant classes of drugs, such as the monoclonal antibodies, is the localization of the target protein in the cells, with cytoplasmic or nuclear proteins that are difficult to be reached and targeted by this approach [14]. Moreover, the mRNA itself could retain the oncological functions before it is translated in the protein. For instance, it was demonstrated that the mRNA could act as a sponge, by recruiting and so neutralizing the other regulatory elements (mostly microRNAs) critical for the maintenance of normal function in the cell, as reported for the MYCN mRNA to the let-7 microRNA [14,15]; therefore, if the drug acts only at the level of the protein, this oncogenic effect would remain, while when the drug acts at the mRNA level, it has the double advantage of blocking the translation into protein and blocking the specific pro-tumor activities of the mRNA.

Finally, in the recent years emerged a large part of the genome containing genes whose transcription generate a large family of different non-coding RNAs (ncRNAs). This ncRNAs plays key regulatory roles in both normal cellular activity and disease, including cancer, but because ncRNAs are not translated into proteins, they could not be targeted by conventional drug discovery approaches directed against the proteins [16,17]. In this context, nucleic acid therapeutics by oligonucleotides that target RNA or DNA sequences, are emerging as a new and highly promising class of drugs, able to tackle the limitations described above, and to open a new era of tailored drugs and precision medicine also in cancer therapy [18,19,20]. Indeed, oligonucleotides showed the ability to also specifically target the so-called undruggable proteins and ncRNAs. Currently, oligonucleotide therapeutics have been only approved for the treatment of different rare diseases [21], but none is yet available for cancer therapy. However, many clinical trials with oligonucleotides are ongoing for tumor treatment, and, in the next few years, some of them are expected to reach the market.

## 2. Oligonucleotide Therapeutics as New Targeted Anti-Cancer Drugs for Challenging or Undruggable Proteins

The molecular targets for therapy could be divided in two major categories, namely, druggable and undruggable. “Druggability” implies that the target molecule must have structures that should allocate the specific binding and inhibition by low-molecular-weight compounds. Typically, a protein is considered druggable if it contains a cavity, usually a well-defined catalytic cleft. It is estimated that almost 70% of the human proteins [22] are considered difficult to be targeted, including the transcription factors that are widely thought to be undruggable due to the lack of catalytic clefts and the much-sought drug-binding pockets. To date, targeting the relevant transcription factors with small molecular compounds remains challenging. In particular, relevant examples are the members of the MYC family and RAS family of oncogenes, that account for amongst the highest rate of mutations in cancers and define aggressive tumor behaviors [23,24]. The strategies that propose the indirect blocking of transcription factors by targeting their upstream or downstream pathway genes could result in less efficacy and also specific side effects in healthy cells. New strategies are urgently needed to generate drugs able to tackle the undruggable targets in cancer. In this respect, oligonucleotide therapeutics enable a direct targeting of the gene by acting at the level of the RNA or at the level of the DNA, based on the Watson-Crick complementary rule of binding. Different classes of oligonucleotides have been developed since the first use in clinical was proposed in the 1970s, but, in particular, antisense oligonucleotides (ASOs), small interference RNAs (siRNAs), and microRNAs (miRNAs) imposed their presence as the most representative for clinical development and therapeutic application [25,26]. For a better understanding of the impact and the relevance of each group, we performed a research study in clinicaltrials.gov (accessed on 1 May 2022) using “Cancer” as the keyword and focusing our attention on clinical trials available for each class of oligonucleotide in analysis.

As expected, the ASOs are the most represented oligonucleotides, accounting for 66% of all of the compounds analyzed (75% of clinical trials), and are the only class with some compounds in clinical phase II or III (Figure 1 and Table 1). Interestingly, only 12% of the clinical studies are recruiting or active, and the majority are completed (59%), or in other status (29%). From a structural point of view, the ASOs range in size from 12 to 30 nucleotides, are single stranded, and work through the classic Watson-Crick base pairing and can act as both gene expression inhibitors or splicing modulators [27,28].

The second most common oligonucleotide class is represented by the siRNAs, accounting for 25% of the compounds. The analysis shows how they are the subject of almost the same number of clinical trials as the miRNAs, but with a higher number of trials in phase I (76% of total siRNAs’ clinical trials). In relation to the clinical trial status, the siRNAs show 35% active/recruiting trials, and 18% of the trials terminated with no suspension (Figure 1 and Table 1). The siRNAs are the longest oligonucleotides in size (20–25 nt), they are double stranded, and work in complex with RISC to post-transcriptionally silence the target gene expressions [29]. These compounds can be chemically synthesized, maintaining the characteristics needed for the proper activation of the enzymatic mRNA degradation, allowing their use as therapeutic compounds [30].

Finally, the miRNAs represent the smallest class with only four compounds in clinical trial (9%). Despite their number, these compounds are well represented in clinical trials, as mentioned before, and are equally distributed among phases 1 and 2. Furthermore, a relevant number of clinical trials are recruiting (17%), while others are now terminated (50%), or withdrawn (17%) (Figure 1 and Table 1). The miRNAs are small (18–25 nt) single stranded non-coding RNAs, containing usually sequences complementary to one or more of the target RNAs [31]. They work similarly to the siRNAs, activating the RISC complex after formation of miRNAs duplex [32]. The miRNAs can also be synthesized by mimicking their normal biological function, to be redirected against specific targets for therapeutic purposes [33].

### 2.1. Oligonucleotide Therapeutics Targeting the MYC Gene Family

The MYC family is composed of c-MYC, MYCN, and MYCL [34] The MYC gene family encodes for the basic helix-loop-helix-leucine zipper (bHLH-LZ) transcription factor proteins which exhibit a high-structural homology, including highly conserved Myc boxes (MB) and a basic region (BR), helix-loop-helix (HLH) and leucin zipper (LZ) motifs [35,36]. The MYC gene family is highly involved in tumors in which they are often dysregulated and mutated (manly by translocations or gene amplification), resulting in overexpression and association with tumor aggressiveness and poor prognosis [37]. The proteins that originate from the MYC gene family are mainly considered undruggable with the conventional approaches that rely on small molecules.

#### 2.1.1. Oligonucleotide Therapeutics Targeting MYC

MYC has been proposed as an important oncogenic target for its role in cell proliferation and survival, angiogenesis, metastasis, drug resistance, and poor patient prognosis [38,39]. MYC-targeted oligonucleotide therapeutics, based on a small interfering RNA lipid-based nanoparticle (DCR-MYC, Dicerna Pharmaceuticals), to inhibit the oncogene MYC at the level of the mRNA, was developed to treat various cancer types, including hepatocellular carcinoma (HCC), solid tumors, lymphoma, or multiple myeloma. The liposomal delivery system of the DCR-MYC is based on EnCore Dicerna’s proprietary technology, due to its specific Envelope and Core lipid contents. For targeting MYC, Dicer-substrate small interfering RNA (DsiRNA) was used in the drug formulation. DsiRNAs, longer duplex RNAs, are Dicer substrates to be subsequently processed into small interfering RNAs (siRNAs), and have an increased potency in RNA interfering processing [40]. The data from the phase I—dose-escalation study indicated that DCR-MYC presents good clinical and metabolic responses in patients at a variety of dose levels [41]. However, phase I (clinicaltrials.gov (accessed on 1 May 2022) NCT02110563) and phase Ib/2 trials (clinicaltrials.gov (accessed on 1 May 2022) NCT02314052) were terminated on the sponsor’s decision, due to a lack of the gene-silencing effectiveness that was anticipated by the company [42].

#### 2.1.2. Oligonucleotide Therapeutics Targeting MYCN

The MYCN oncogene is a well-known driver of different, highly aggressive tumors (including Neuroblastoma, Small-Cell Lung Cancer, Rhabdomyosarcoma), where it is dysregulated and amplified and is strongly associated with poor survival prognosis [43,44]. MYCN overexpression reprograms the tumor cells towards a stem-like phenotype that promotes proliferation and cell growth, while inhibiting cell differentiation and apoptosis. It also favors immune escape, invasion, metastases, and angiogenesis [45,46]. Interestingly, MYCN is expressed during embryogenesis and has a highly restricted pattern of expression in normal cells after birth [47]. All of these factors make the N-Myc protein a promising target for a tumor-specific therapy. However, inhibitors against the N-Myc protein have, to-date, largely failed and have led to N-Myc being currently considered to be an undruggable target [48].

It has been demonstrated that an alternative approach concerns specific gene expression inhibition at the level of DNA through a MYCN-specific antigene peptide nucleic acid (agPNA) oligonucleotide [49,50]. The antigene oligonucleotide approach (via persistent blocking at the level of transcription) has shown advantages in blocking translation by the antisense oligonucleotide strategies. The peptide nucleic acids (PNAs) have shown promising results as antigenes, due to their resistance to proteases and nucleases and their ability to potently and specifically bind the target DNA [51,52]. Differing from the use of antisense oligonucleotides, which inhibit mRNA translation, the antigene approach involves binding to the chromosomal DNA, resulting in the inhibition of transcription. By persistently blocking the transcription, the antigene oligonucleotides showed higher efficacy compared with antisense oligonucleotides [49,51]. Antigene therapy by targeting MYCN transcription has great potential in treating MYCN-expressing tumors, as was previously demonstrated in the preclinical treatment of neuroblastoma and rhabdomyosarcoma by MYCN-specific agPNA [50,51]. Neuroblastoma (NB) is the deadliest pediatric tumor. Approximately 25% of patients with a NB diagnosis present with MYCN amplification (MNA), which is linked to a poor prognosis, metastasis, and recurrence [53,54]. It has been shown that BGA002, a new and highly improved agPNA oligonucleotide, is able to specifically target a unique sequence on the human MYCN gene [55]. BGA002 showed a specific, dose-dependent decrease in the MYCN mRNA and protein, while decreasing the viability in a panel of 20 NB cell lines, followed by the block of different MYCN tumorigenic alterations, and to the anti-tumor efficacy of BGA002 in vivo in a MNA NB mouse model [55]. Moreover, while MYCN drives a tumor immunosuppressive environment, which impacts survival in several MYCN-positive tumors, the block of MYCN by the anti-MYCN BGA002 is able to reactivate and restore the effectiveness of the natural killer immune cells against NB [56]. It has been also found that BGA002 restores the retinoic acid (RA) response, leading to a differentiation or apoptosis in the MNA NB and also to a significant increase in survival in a mouse model of MNA-NB [57]. This study shows that it is possible to realize precision medicine by the identification of optimal combined drugs that can achieve a potent and selective block of cancer pathways only in tumor cells, preserving the impact of side effects on normal cells. MNA is not restricted to NB, and the restoration of RA treatment could be beneficial in different MNA tumors. BGA002 has received orphan drug designation from the Food and Drug Administration (orphan registry: DRU-2017-6085) and from the European Medicines Agency (orphan registry: EU/3/12/1016). Based upon its well-tolerated regulatory safety profile package, BGA002 is now moving to phase I clinical trials in neuroblastoma patients.

### 2.2. Oligonucleotide Therapeutics Targeting the RAS Gene Family

Ras proteins regulate the activation of different patterns strongly involved in cancer, such as cell proliferation, differentiation, and survival [58]. This family of proteins is encoded by three ubiquitously expressed genes, HRAS, KRAS, and NRAS, that share most of their sequence and function [59]. In normal conditions, they act as the activator of more than 20 different proteins from different effectors’ families [60], so their constitutive activation may cause a deregulation of many cell functions and lead to cancer. In particular, the frequency and the pattern of the gene mutations can associate different Ras genes with different cancer types [61]. While each Ras member is involved in cancer, KRAS is surely the major cancer-causing isoform, accounting for 75% of all Ras-associated tumors [62]. The other isoforms NRAS (17%) and HRAS (7%) account for only a small subset of cancer types [62]. In the past years, many attempts were performed to develop direct Ras gene family inhibitors, but the protein structure showed characteristics that were not very compatible with the small molecules’ approach [63]. As the same post-translational approach was ineffective, due to isoform-specific differences [64], so a more specific approach was needed in consideration of the relevance of the KRAS specific isoform to the others. In particular, the oligonucleotides have a promising therapeutic potential as mutant-specific RAS inhibitors, active against any major mutation.

#### Oligonucleotide Therapeutics Targeting KRAS

The KRAS-targeted siRNA-polymeric nanoparticles for local therapy, siG12D-LODER, were designed by Silenseed Ltd. for patients with locally advanced pancreatic cancer. The biodegradable polymer matrix, Local Drug EluteR (LODER), was used to release the G12D-mutated KRAS-targeted siRNA locally within a pancreatic tumor microenvironment for controlled and prolonged delivery [65]. The LODER matrix consists of a copolymer of poly lactic-*co*-glycolic acid (PLGA) of a high molecular weight greater than 50 kD. The siG12D-LODER was designed to be properly inserted and placed into the tumor using a standard biopsy procedure [66,67]. A slow and stable release of siRNA from siG12D-LODER over a few months was demonstrated when incubated in PBS. The siG12D-LODER remarkably suppressed the growth of pancreatic tumors in both subcutaneous and orthotopic, xenograft, and syngraft mouse models, without causing any toxicity [65]. In another preclinical study, following subcutaneous implantation of the siG12D-LODER, all of the rats exhibited local and systemic safety and tolerability, without any adverse effects or deaths [67]. A phase I clinical trial was conducted by injection of the siG12D-LODER drug into patients via the endoscopic ultrasound (EUS) biopsy needle (clinicaltrials.gov (accessed on 1 May 2022) NCT01188785). The patients received a combination treatment of siG12D-LODER with gemcitabine or FOLFIRINOX in the phase II study, reporting an enhanced therapeutic effect [66]. Phase II trials of siG12D-LODER with gemcitabine + nab-paclitaxel are currently underway (clinicaltrials.gov (accessed on 1 May 2022) NCT01676259).

The ASO strategy has also been proposed for the treatment of the KRAS mutation in cancer. AZD4785 is a cEt-modified ASO [68], complementary to a KRAS mRNA sequence, developed by Ionis in collaboration with Astra Zeneca. Interestingly, the advanced chemistry of this compound and the resulting potency, allowed its use without any delivery agent in the first preclinical studies. As expected, AZD4785 is able to directly downregulate KRAS mRNA at the nM level in vitro (IC_50_ 10 nM), and is also able to increase survival and reduce tumor growth in a mouse model of lung cancer [69]. From the clinical point of view, only one phase I study is reported (NCT03101839) as completed in 2017, without any recent advancement.

### 2.3. Oligonucleotide Therapeutics Targeting STAT3

STAT3 is a protein activated by members of the JAK family through phosphorylation [70]. In its phosphorylated form, STAT3 dimerizes and works in the nucleus as a transcription factor involved in cell proliferation, development, differentiation, inflammation, and apoptosis. Constitutive activation can be found in several types of human cancer [71,72], and it is able to increase the level of different cancer-related molecules, such as surviving, Bcl-XL, cyclin D1/D2, C-Myc, Mcl-1, and vascular endothelial growth factor (VEGF), favoring tumorigenic progression [73,74]. For this reason, STAT3 represents one of the most interesting targets for therapeutics in oncology, and in the past years many strategies were developed to effectively inhibit STAT3 expression. For example, synthetic inhibitors, such as CDDO-Me or FLLL32, can reduce the activity of the protein downstream, blocking the JAK/STAT3 interaction or inhibiting the DNA binding process but cannot overcome the issue related to the overall aspect of STAT3 overexpression [75,76]. In a different way, oligonucleotide compounds are able to specifically target STAT3 reducing the protein expression, directly downregulating the mRNA. A new compound, such as AZD9150 (ISIS 481464), a 16-nucleotide next generation chemistry antisense oligonucleotide [77], or CpG-Stat3 siRNA, a conjugate of an oligonucleotide TLR9 agonist linked to a STAT3 siRNA [78], are, in fact, designed to specifically reach this aim. In particular, the combined use of synthetic oligonucleotide agonists for TLRs and siRNA target-specific, is a novel and potent strategy that can achieve both target delivery enhancement of siRNA to immune cells and antitumor immune response activation [78]. Furthermore, this strategy can target a broad spectrum of TLRs, including TLR3, TLR7, TLR8 and TLR9, allowing the selection of the best combination of oligonucleotides and receptors [79,80]. The TLR9-specific oligodeoxynucleotides, containing an unmethylated CpG-motif (CpG ODN), represent the first choice, because they are already in clinical testing [81]. Additionally, CpG ODN are efficiently internalized by various antigene-presenting cells (APC), such as macrophages, B cells, and dendritic cells (DCs), and their binding to TLR-9 can initiate immune response cascade, effectively providing the immuno-stimulation [81,82]. Obviously, the interest in these new compounds is high, and preclinical phase I and/or 2 clinical studies are still ongoing and promising.

### 2.4. Oligonucleotide Therapeutics Targeting BCL-2

BCL-2 is one of the first-discovered regulators of the apoptosis process. This gene is overexpressed by the translocation *t*(14;18) in B-cell lymphoma and is implicated in many different cancers, such as melanoma, breast, and lung carcinoma [83]. BCL-2 is not only involved in the neoplastic development, but also in the resistance mechanism to cancer treatment [84]. In this context, the therapies pointed at BCL-2 inhibition can be considered crucial to overcome resistance to the common strategies for cancer treatments [85]. Oblimersen (G3139) is certainly the most studied BCL-2 inhibitor and was involved in many clinical studies (Table 1). Unfortunately, despite its high efficacy in vitro and in vivo preclinical studies [86,87], the clinical studies showed how it is important to administer this compound in combination to achieve significant results. Phase 3 clinical studies (data not shown) are indeed available, but only for this form of application. It is important to underline that other BCL-2 inhibitors have been developed, such as BP1002 [88] and PNT2258 [89]. Both of them are still in clinical phase I trials; the first active, and the second successfully completed, and they represent promising compounds for the future of the Bcl-2 inhibitor.

## 3. Oligonucleotide Therapeutics as New Targeted Anti-Cancer Drugs for Non-Coding RNAs

The only function of RNA in cells was thought to be as a template for protein synthesis, but extensive research in the last few decades proved otherwise [90]. For many years, scientists have called the non-protein-coding part of the human genome as “junk DNA”. Approximately 80% of the genome is biologically active and transcribed to RNA, while only 2% is transcribed to protein-coding mRNA [91], as groundbreaking projects on the human genome, such as the FANTOM [92] and ENCODE [93], have contributed, revealed and elucidated. Non-coding RNAs (ncRNAs) constitute a large part of the genome and are transcribed from DNA but not translated into protein [94]. The ncRNAs can be classified in consideration of their function and/or dimension in many different ways [95,96]. Normally, ncRNA are conventionally divided into two major groups, based on the threshold of 200 nucleotides (nt) of length, namely short non-coding RNA (sncRNA) and long non-coding RNA (lncRNA) [90,97]. The sncRNAs (<200 nt) include both structural and regulatory RNA [97,98]. The structural sncRNAs include the RNAs involved in fundamental housekeeping functions. For example, the ribosomal RNAs (rRNA) and transfer RNAs (tRNAs) are involved in the mRNA translation process to protein, while the small nuclear RNAs (snRNAs) and small nucleolar RNAs (snoRNAs) resemble other fundamental functions concerning rRNA modification and/or mRNA maturation [95,99]. In addition to these conventional RNAs, the sncRNAs include several regulatory RNAs, such as microRNAs (miRNAs), small interference RNAs (siRNAs) and Piwi-interacting RNAs (piRNAs), all of which are able to regulate gene expression through different mechanisms [100,101]. The LncRNAs (>200 nt) are a group of more heterogeneous ncRNAs, with an unknown function. Interestingly, the lncRNAs show many characteristics in common with the coding RNA, having the possibility of expressing differentially, using splice variants [102], and including many different types of RNA with a great variety of mechanisms of action, mostly used to term and classify it [103].

The expression of some ncRNAs is dysregulated and contributes to the development and progression of various diseases, including cancer [104]. The ncRNAs have been shown to have great potential as therapeutic targets, drugs, and diagnostic markers [105,106]. Therefore, the ncRNA-based therapies by oligonucleotides have gained great interest for use with the targets considered “undruggable”, and for their potential use as a part of precision or personalized medicine in the treatment of a wide range of diseases, including cancer [107]. With the growing body of evidence on the biology and clinical significance of ncRNA, a number of investigators in academia, biotechnology, and pharmaceutical companies are currently working on developing oligonucleotide therapeutics to regulate the pathogenic ncRNA gene expression in various diseases [105,108]. miRNA and siRNA are the most studied ncRNAs and, over the past decade, 23 clinical trials (Figure 1 and Table 1) have been initiated for testing the efficacy of miRNA and siRNA in cancer therapeutics. Several miRNA-based therapeutics recently moved to phase II clinical trials for advanced cancers, including TargomiR (miR-16 mimic-based therapy) in mesothelioma [109], Cobomarsen (anti-miR-155) in T-cell leukemia/lymphoma [110], and Miravirsen (anti-miR-122) in individuals with hepatitis C infection [111].

### 3.1. microRNA Therapeutics

Two types of oligonucleotides are currently being used as miRNA therapeutics: miRNA mimetics to restore the levels of miRNAs downregulated in cancer; and antagomiRs to inhibit overexpressed miRNAs [112]. Thus far, six clinical trials have been investigating miRNA therapeutics’ potential as oncology drugs. Of these, three were terminated and one withdrawn, leaving one phase I trial completed and one still recruiting patients. The termination of NCT01829971 and the withdrawal of NCT02862145 were due to frequently related serious adverse events reported by the investigators, while the NCT03837457 and NCT03713320 trials were terminated due to business reasons and patients placed in other trials, according to the comments on clinicaltrials.gov (accessed on 1 May 2022) [113].

#### 3.1.1. Cobomarsen

Cobomarsen, or MRG-106 (anti-miR-155, miRagen Therapeutics, Inc, Boulder, Colorado), is a locked nucleic acid (LNA)-modified oligonucleotide inhibitor of miR-155. miR-155 is found at high levels and is related to a worse prognosis in lymphoma and leukemia. It plays an important role in the progression of mycosis fungoides (MF), the most common type of cutaneous T-cell lymphoma (CTCL) [114]. The preclinical studies of cobomarsen in NSG mice carrying B-cell lymphoma (ABC-DLBCL) xenografts indicated that by intravenous injection, the tumor volume was reduced, apoptosis was induced, and the expression of direct target of miR-155 was upregulated. In addition, it was emphasized that the drug did not exhibit any toxic effects [115]. Phase I clinical studies evaluated the safety, tolerability, pharmacokinetics, and potential efficacy of the drug in patients with a subset of lymphomas and leukemias. The patients received the drug by subcutaneous or intravenous injection for six dosages in the first 26 days, later once a week. The results showed that the treatments sustained a reduction in the lesion burden accompanied by an inhibited miR-155 level. Moreover, it displayed an efficient clinical activity without serious adverse effects (clinicaltrials.gov (accessed on 1 May 2022) NCT02706886) [116]. Phase II clinical trials began to compare the efficacy and safety of cobomarsen with vorinostat, the Food and Drug Administration (FDA) approved the drug for CTCL (clinicaltrials.gov (accessed on 1 May 2022) NCT03713320), and to study its efficacy and safety in subjects who have confirmed disease progression after treatment with vorinostat (clinicaltrials.gov (accessed on 1 May 2022) NCT03837457) [117]. However, phase II studies were terminated on the company’s decision, without any safety or efficacy issues [118]. In the phase I trial (NCT02580552), the patients showed acceptable toxicity and drug responses in all four of the patients recruited for the intra-tumoral injection, with decreased neoplastic cell density and depth [110]. The FDA granted Orphan Drug Designation to cobomarsen for mycosis fungoides type cutaneous T-cell lymphoma in 2017.

#### 3.1.2. TargomiRs (a miR-16 Mimic and a MIR16-Based miRNA Mimetic)

TargomiRs is a miRNA replacement therapy and is the first technology to complete phase I trials of carrier-based miRNA therapeutics in cancer patients [119]. It consists of three components: (I) a miR-16 mimic that acts as a tumor suppressor in a range of cancer types; (II) an EnGeneIC Delivery Vehicle (EDV); and (III) a targeting ligand; anti-epidermal growth factor (EGFR) bispecific antibody. EDVs are non-living bacterial minicells of a 400 nm diameter that have the ability to deliver chemotherapeutic drugs, nucleic acids, and proteins [120]. Bcl-2 and CCND1 genes, which promote cancer progression, have been reported to be important targets of miR-16. In vitro studies indicate that depleted miR-16 levels could be restored with MIR16 mimetics in malignant pleural mesothelioma cell lines, which led to growth inhibition [121]. In vivo delivery of the miR-16 mimic via EDV to nude mice with malignant pleural mesothelioma (MPM) xenografted tumors significantly inhibited the tumor growth without any safety concerns from the minicells. Additionally, the miR-16 mimic, at doses less than 1 lg, had a powerful tumor inhibitory effect in vivo [121]. An open-label, dose-escalation phase I study of TargomiRs (also called MesomiR-1) was conducted in patients with MPM. The results demonstrated that TargomiRs was well tolerated, together with early signs of antitumor efficacy and encouraging survival in patients. In addition, the phase II trial, in which TargomiRs will be added to standard chemotherapy, will begin soon (clinicaltrials.gov (accessed on 1 May 2022) NCT02369198) [109].

#### 3.1.3. MRX34 (MiR-34a Mimic)

Phase I trials of MRX34, a MiR-34a mimic, were initiated in 2013 in patients with multiple solid tumors, including primary liver cancer, renal cell carcinoma, multiple myeloma, lymphoma, or small-cell lung cancer, with the strategy of restoring endogenous miR-34 using miRNA mimics. The liposome formulation is comprised of amphoteric lipids which are anionic at neutral or higher pH and cationic at lower pH values. This technology facilitates liposome formation, mixing lipids and miRNA in an acid environment, and increasing the specific tumor targeting. In fact, in normal biological fluids (at pH7–7.5), the nanoparticles may prevent an unwanted interaction with the negative charge of cellular membrane, but at the same time they can improve adhesion to tumor cells, in consideration of the lower pH normally found in tumor areas [122]. MRX34 was reported to directly regulate 24 oncogenes carrying essential roles in proliferation, cell cycle, metastasis, anti-apoptosis, the cancer cell self-renewal process, oncogenic transcription, and chemoresistance [123]. The systemic delivery of MRX34 into mice with orthotopic Hep3B and HuH7 liver cancer xenografts resulted in dramatic tumor growth reduction, and even tumor regression. The therapeutic doses used had no harmful side effects or immune stimulation in mice. Following a single administration of MRX34, multiple oncogenes of the key cancer pathways were inhibited, including WnT/b-Catenin, MapK, c-Met, Hedgehog, and vascular endothelial growth factor (VEGF), while multiple genes of the p53 pathway were stimulated [124]. However, despite its very high therapeutic performance in preclinical studies, in 2016, Mirna Therapeutics discontinued the phase I trials, due to immune-related severe adverse effects (SAEs) observed in five patients that resulted in four of their deaths (clinicaltrials.gov (accessed on 1 May 2022) NCT01829971). Further phase I and II studies of MRX34 for melanoma treatment were also withdrawn because of SAE (clinicaltrials.gov (accessed on 1 May 2022) NCT02862145). Nevertheless, a recently published study reported that MRX34 treatment with dexamethasone premedication exhibited clinical activity and a manageable toxicity profile in most of the patients [125]. All of the miRNA therapeutics across indications are still in clinical trials, with some positive outcomes thus far. Studies on miRNAs for cancer diagnosis have also exponentially increased in the last few years, and miRNA-based diagnostic tools are being developed [112].

## 4. Challenges and Solutions for the Clinical Success of Oligonucleotide Therapeutics 

The pharmacokinetic properties of oligonucleotides have been extensively characterized. Oligonucleotides show a wide tissue biodistribution in several organs, and the kidney, liver and spleen are the principal target organs [126,127]. In general, oligonucleotides are negatively charged and this chemical characteristic limits the cross of the blood-brain barrier, and consequently the localization to the brain and spinal cord. Thus, the treatment of brain tumors or of brain metastases of tumors that arise in other organs is not feasible by systemic treatment, but should require a route of administration that delivers the drug into the central nervous system [128]. Another possibility is the use of delivery systems, such as liposomes that could cross the blood-brain barrier [129]. The phosphorothioate modification of the backbone greatly improved the binding of the oligonucleotides to the plasma proteins, increasing the time of the oligonucleotides in the circulation, while reducing their elimination. In vivo stability and resistance to degradation by DNases or RNases have been greatly improved by the different chemical modifications introduced in the nucleotides and in the backbone [130,131]. Regarding the delivery to the target cells, oligonucleotides are hydrophilic molecules and cannot passively cross the plasma membrane in the same way as the small molecule lipophilic drugs. The delivery of oligonucleotides is mainly mediated by the formation of endosomes, but then they should be released to the cytoplasm before the fusion of the endosome with the lysosome triggers their degradation. In this respect, several systems of oligonucleotide delivery have focused on their endosomal escape to avoid their degradation [132,133,134]. Oligonucleotides are used without the delivery systems, or formulated with liposomes, nanoparticles, or conjugated with carriers, such as peptides, small molecules, and cell surface receptors. These delivery systems confer improved pharmacokinetic and pharmacodynamic in vivo properties, but should be carefully evaluated for their potential side effects, and the overall therapeutic index that should respect the means of the oligonucleotides when used, without any carrier. Many tumors are highly vascularized, and several reports indicate that cancer cells can be accessible and targeted by oligonucleotides [135,136,137].

Regarding the safety profile of the oligonucleotides, these could be related to the potential inhibition of the expression of off-target genes. However, performing a robust process of bioinformatic prediction of the complementary target sequences, and by the selection of optimal unique sequences only present in the target gene, followed by the experimental evaluation of potential off-target genes, means that it is possible to identify potent and specific drug-candidate oligonucleotides that maximize the on-target effects and greatly minimize the potential for off-target effects at a level that do not impact healthy cells [138,139,140]. Moreover, the introduction of the different chemical modifications to the oligonucleotide is reported to confer different improved characteristics. First generation ASOs, for example, are able to efficiently recruit RNase H (methylphosphonate and phosphorothioate); however, they show limited clinical application due to their sensibility for endo- and exonucleases as well as their limited binding affinity for the target and potential cytotoxic side effects [141,142]. Second generation ASOs introduce the first important improvement in oligonucleotides in overall stability and target engagement, reducing at the same time the off-target toxic effects [143,144]. Finally, the third generation ASOs shows the most enhanced resistance against nucleases and the best target affinity, compared to the previous generations [144,145,146]. The last generation is also potentially considered the most safe, thanks to its high specificity, affinity, and very limited off-target effects that can be avoided with many bioinformatic tools [131]. Thus, the chemical modification introduced into DNA or RNA oligonucleotides (Figure 2) have allowed an improvement in the specificity, stability, safety and pharmacokinetic/pharmacodynamic profiles, making this class of compounds more reliable for clinical purposes [131,144]. Other potential toxic effects of oligonucleotides that are reported in preclinical and clinical studies could be related to a stimulation of the immune system [147], liver inflammation, or accumulation in the proximal tubule of the kidney [148,149,150]. However, these phenomena are usually not serious events and could be recovered after cessation of the oligonucleotide administration.

Regarding the potential stimulation of the immune system, in DNA-based oligonucleotides this phenomenon is related to the presence of CGs in the oligonucleotide that are recognized by the TLR9, followed by cytokine release and activation of the innate immunity [151,152]. In RNA-based oligonucleotides, the immune system stimulation is related to the activation of TLR3 and TLR7 [152,153]. Nevertheless, in the context of a cancer therapy, the activation of the immune system could not necessary be considered a side effect, but rather an advantage and a potential dual anti-tumor effect of the oligonucleotide, in addition to the on-target inhibition of its specific cancer gene. 

## 5. Challenges and Solutions Related to the Oligonucleotides for Cancer Therapy

Because of the high rate of new mutations that occur in the tumors, the new targeted drugs based on small molecules could also result in being ineffective, especially if the mutations fall in the functional domains (most often catalytic domains) of the proteins that are the most relevant target sites of these class of drugs [154,155]. For these reasons, new targeted cancer therapeutics are needed to enhance the efficacy and reduce the side effects. For a successful use of oligonucleotides for cancer therapy, many critical aspects and challenges have to be considered and solved. An optimal cancer treatment should employ a sustained therapeutic schedule, able to counteract the continuous growth of the tumor, especially those with aggressive characteristics. The most relevant and causative target genes in cancer present the challenge that they are often overexpressed. Therefore, to reach a therapeutic response, the oligonucleotide should exert a pharmacological inhibitory activity by contrasting its target gene, that is often constitutively activated at a high level of expression. However, a relevant level of inhibition of the target gene expression is not always required, because the cancer cells, following the downregulation of the critical key oncogenes, could become vulnerable and susceptible to elimination by other drugs or by the immune system, even if high levels of targeted gene expression inhibition are not reached.

## 6. Conclusions and Future Perspectives

Oligonucleotide therapeutics appear particularly adequate to respond to the challenges mentioned above. They could also expand the range of targets for a specific therapy for challenging or undruggable targets. Moreover, the drug discovery process to identify a new lead candidate oligonucleotide drug is, in general, shorter than the time required to identify a conventional lead drug based on a small molecule compound. This aspect is relevant to adapting and responding in a specific manner to the high rate of tumor mutations and heterogeneity. The numerous advantages offered by the oligonucleotide position them as an emerging and promising class of biotherapeutics for a new era of precision and personalized medicine in cancer therapy.

## Figures and Tables

**Figure 1 pharmaceutics-14-01453-f001:**
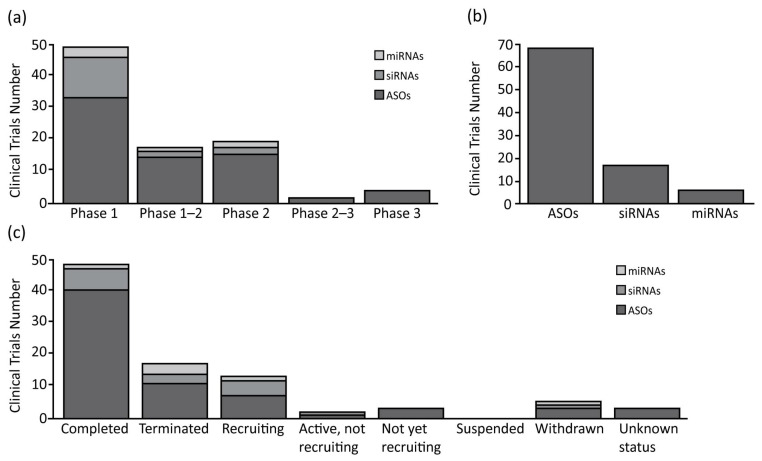
Summary of clinical trials therapies using oligonucleotide in oncology. (**a**) Antisense oligonucleotides (ASOs), small interfering RNAs (siRNAs) and microRNAs (miRNAs) number of clinical trials divided for clinical phases; (**b**) Total clinical trials number for each class of oligonucleotide analyzed; (**c**) ASOs, siRNAs and miRNAs number of clinical trials divided for clinical status condition.

**Figure 2 pharmaceutics-14-01453-f002:**
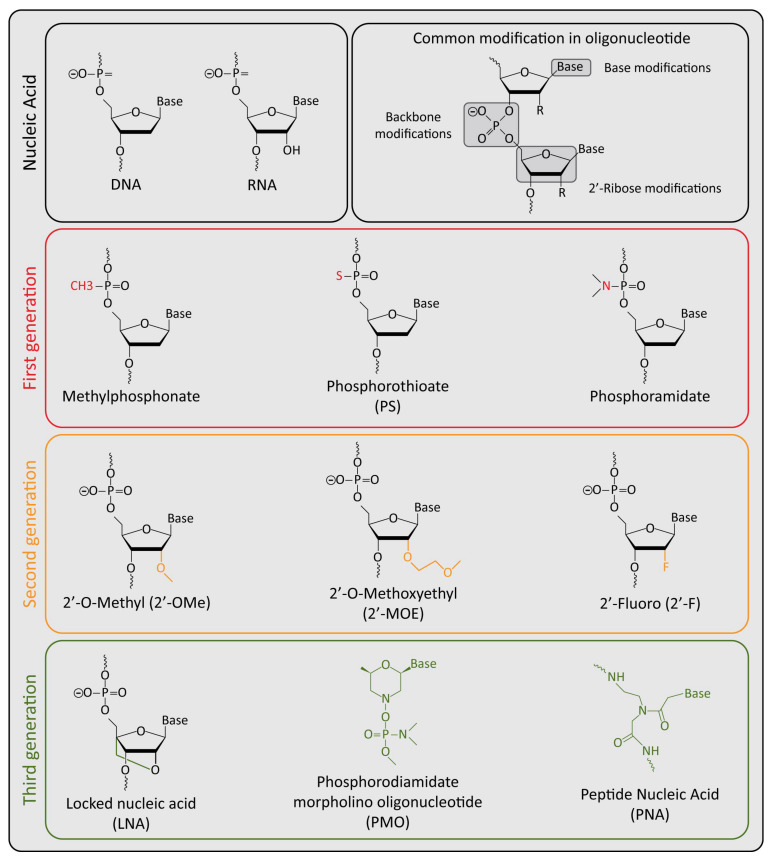
Graphic representation of chemical specific characteristics for each generation ASOs. Main chemical differences from common shared structure are marked with colors.

**Table 1 pharmaceutics-14-01453-t001:** Summary of oligonucleotides in clinical trials for oncology.

Oligonucleotide	Target	Drug Type	Cancer Type	Clinical Phase	Clinical Trials ID
WGI-0301	AKT1	ASO	Advanced Solid Tumors	Phase 1	NCT05267899
AZD5312	AR	ASO	Advanced Solid Tumors with Androgen Receptor Pathway as a Potential Factor	Phase 1	NCT02144051
BP1002 (L-Bcl-2 antisense oligonucleotide)	BCL-2	ASO	Mantle Cell Lymphoma|Peripheral T-cell Lymphoma (PTCL)|Cutaneous T-cell Lymphoma (CTCL)|Chronic Lymphocytic Leukemia (CLL)|Small Lymphocytic Lymphoma (SLL)|Follicular Lymphoma|Marginal Zone Lymphoma|Hodgkin Lymphoma|Waldenstrom Macroglobulinemia|DLBCL	Phase 1	NCT04072458
PNT2258	BCL-2	ASO	Cancer|Lymphoma|Prostate Cancer|Melanoma	Phase 1	NCT01191775
LErafAON	CRAF	ASO	Neoplasms	Phase 1	NCT00024648|NCT00024661|NCT00100672
AZD8701	FOXP3	ASO	Clear Cell Renal Cell Cancer|Non-Small-Cell Lung Cancer|Triple Negative Breast Neoplasms|Squamous Cell Cancer of Head and Neck|Small-Cell Lung Cancer|Gastroesophageal Cancer|Melanoma|Cervical Cancer|Advanced Solid Tumors	Phase 1	NCT04504669
BP1001-A (Liposomal Grb2 Antisense Oligonucleotide)	GRB-2	ASO	Solid Tumor, Adult|Carcinoma, Ovarian Epithelial|Fallopian Tube Neoplasms|Endometrial Cancer|Peritoneal Cancer|Solid Tumor	Phase 1	NCT04196257
EZN-2968	HIF-1α	ASO	Neoplasms|Liver Metastases|Carcinoma|Lymphoma	Phase 1	NCT01120288|NCT00466583
AZD4785	KRAS	ASO	Non-Small-Cell Lung Cancer|Advanced Solid Tumors	Phase 1	NCT03101839
TASO-001	TGF-β2	ASO	Solid Tumor	Phase 1	NCT04862767
SD-101 (CpG Oligonucleotide)	TLR9	ASO	Advanced Malignant Solid Neoplasm|Extracranial Solid Neoplasm|Metastatic Malignant Solid Neoplasm	Phase 1	NCT03831295
CpG7909 (PF3512676)	TLR9	ASO	Intraocular Melanoma|Malignant Conjunctival Neoplasm|Melanoma (Skin)	Phase 1	NCT00471471
ISS 1018 (CpG ODN)	TLR9	ASO	Colorectal Neoplasms	Phase 1	NCT00403052
IONIS-STAT3Rx (AZD9150)	STAT3	ASO	Hepatocellular Carcinoma|Ovarian Cancer|Ascites|Gastrointestinal Cancer|Advanced Cancers|DLBCL|Lymphoma	Phase 1|2	NCT01839604|NCT02417753|NCT01563302|NCT02549651
ISIS 183750	eIF4E	ASO	Colorectal Neoplasms|Colorectal Carcinoma|Colorectal Tumors	Phase 1|Phase 2	NCT01675128
Apatorsen (OGX-427)	HSP-27	ASO	Squamous Cell Lung Cancer|Bladder Cancer|Urothelial Carcinoma|Prostate Cancer	Phase 1|Phase 2	NCT02423590|NCT00959868|NCT00487786|NCT01780545|NCT01120470
GTI-2040	R2 subunit of RNR	ASO	Carcinoma, Renal Cell|Metastases, Neoplasm	Phase 1|Phase 2	NCT00056173
Aezea (Cenersen)	TP53	ASO	Myelodysplastic Syndromes|Acute Myelogenous Leukemia	Phase 1|Phase 2	NCT02243124|NCT00967512
VEGF-Antisense Oligonucleotide	VEGF	ASO	Mesothelioma	Phase 1|Phase 2	NCT00668499
AEG35156	XIAP	ASO	Human Mammary Carcinoma|Carcinoma|Pancreas|Non-Small-Cell Lung	Phase 1|Phase 2	NCT00385775|NCT00558545|NCT00557596|NCT00558922
XIAP Antisense	XIAP	ASO	Leukemia, Myelomonocytic, Acute	Phase 1|Phase 2	NCT00363974
Oblimersen (G3139)	BCL-2	ASO	Lymphoma|Prostate Cancer|Lung Cancer|Melanoma (Skin)|Colorectal Cancer|Breast Cancer	Phase 1|Phase 2|Phase 3	NCT00070083|NCT00080847|NCT00017251|NCT00070343|NCT00016263|NCT00017602|NCT00085228|NCT00030641|NCT00063934|NCT00004870|NCT00005032|NCT00054548|NCT00543231|NCT00543205|NCT00636545|NCT00078234|NCT00021749|NCT00024440|NCT00059813
G4460 (c-myb antisense oligonucleotide)	C-MYB	ASO	Hematologic Malignancies	Phase 2	NCT00780052|NCT00002592
ISIS 5132	CRAF	ASO	Breast Cancer	Phase 2	NCT00003236
BP1001 (Liposomal Grb2 Antisense Oligonucleotide)	GRB-2	ASO	Recurrent Adult Acute Myeloid Leukemia|Acute Lymphoblastic Leukemia|Myelodysplastic Syndrome|Ph1 Positive CML	Phase 2	NCT02923986|NCT02781883|NCT01159028
IGV-001 Cell Immunotherapy	IGF type 1 receptor	ASO	Glioblastoma Multiforme|Glioblastoma	Phase 2	NCT04485949
ISIS 3521	PKCα	ASO	Breast Cancer	Phase 2	NCT00003236
STP705	TGF-β1 and COX-2	ASO	Squamous Cell Carcinoma in Situ	Phase 2	NCT04844983
CpG-ODN	TLR9	ASO	Glioblastoma|Lung Cancer|Hepatocellular Carcinoma|Solid Tumor	Phase 2	NCT00190424|NCT04952272
Custirsen (OGX-011)	clusterin	ASO	Prostate Cancer|Bladder Cancer|Breast Cancer|Kidney Cancer|Lung Cancer|Ovarian Cancer|Unspecified Adult Solid Tumor	Phase 2|Phase 3	NCT00054106|NCT00258375|NCT00471432|NCT01083615
INT-1B3	JNK1	miRNA	Solid Tumor	Phase 1	NCT04675996
TargomiRs	Multiple oncogenes, including BCL2, MCL1, CCND1, and WNT3A	miRNA	Malignant Pleural Mesothelioma|Non-Small-Cell Lung Cancer	Phase 1	NCT02369198
MRX34	30 unique oncogenes, including but not limited to MET, MYC, PDGFR-a, CDK4/6 and BCL2	miRNA	Melanoma	Phase 1|Phase 2	NCT01829971|NCT02862145
Cobomarsen (MRG-106)	mir-155	miRNA	Cutaneous T-Cell Lymphoma/Mycosis Fungoides	Phase 2	NCT03837457|NCT03713320
siRNA-transfected peripheral blood mononuclear cells APN401	CBLB	siRNA	Metastatic Malignant Neoplasm in the Brain|Metastatic Solid Neoplasm|Recurrent Colorectal Carcinoma|Recurrent Melanoma|Recurrent Pancreatic Cancer|Recurrent Renal Cell Cancer	Phase 1	NCT03087591|NCT02166255
EphA2-targeting DOPC-encapsulated siRNA	EPHA2	siRNA	Advanced Malignant Solid Neoplasm	Phase 1	NCT01591356
NBF-006	GSTP	siRNA	Non-Small-Cell Lung Cancer|Pancreatic Cancer|Colorectal Cancer	Phase 1	NCT03819387
Mesenchymal Stromal Cells-derived Exosomes with KRAS G12D siRNA	KRASG12D	siRNA	Metastatic Pancreatic Adenocarcinoma|Pancreatic Ductal Adenocarcinoma	Phase 1	NCT03608631
Proteasome siRNA and tumor antigen RNA-transfected dendritic cells	LMP2, LMP7, MECL1	siRNA	Metastatic Melanoma|Absence of CNS Metastases	Phase 1	NCT00672542
CALAA-01	M2 subunit of ribonucleotide reductase (R2)	siRNA	Cancer|Solid Tumor	Phase 1	NCT00689065
TKM-080301	PLK1	siRNA	Colorectal Cancer with Hepatic Metastases|Pancreas Cancer with Hepatic Metastase|Gastric Cancer With Hepatic Metastases|Breast Cancer With Hepatic	Phase 1	NCT01437007
SLN124	TMPRSS6	siRNA	Non-transfusion-dependent Thalassemia|Low Risk Myelodysplastic Syndrome	Phase 1	NCT04176653
DCR-MYC	MYC	siRNA	Solid Tumors|Multiple Myeloma|Non-Hodgkins Lymphoma|Pancreatic Neuroendocrine Tumors|PNET|NHL| Hepatocellular Carcinoma	Phase 1|Phase 2	NCT02110563|NCT02314052
Atu027	PNK3	siRNA	Advanced Solid Tumors|Carcinoma, Pancreatic Ductal	Phase 1|Phase 2	NCT00938574|NCT01808638
siG12D LODER	KRASG12D	siRNA	Pancreatic Ductal Adenocarcinoma|Pancreatic Cancer	Phase 2	NCT01188785|NCT01676259
STP705	TGF-β1, COX-2 mRNA	siRNA	Squamous Cell Carcinoma in Situ	Phase 2	NCT04844983
CpG-STAT3 siRNA CAS3/SS3	TLR9 and STAT3	siRNA	B-Cell Non-Hodgkin Lymphoma|Diffuse Large B-Cell Lymphoma|Follicular Lymphoma|Mantle Cell Lymphoma|Marginal Zone Lymphoma|Small Lymphocytic Lymphoma	Phase 1	NCT04995536

## Data Availability

Publicly available datasets at https://www.clinicaltrials.gov/ were analyzed in this study.

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
