# Peer review of "Precision Anti-Cancer Medicines by Oligonucleotide Therapeutics in Clinical Research Targeting Undruggable Proteins and Non-Coding RNAs"

_pharmaceutics, 2022, doi:10.3390/pharmaceutics14071453_

Round 1
Reviewer 1 Report
In this manuscript, the authors summarized and reviewed the UpToDate clinical trials of oligonucleotide therapeutics for anticancer purposes. The authors dissected this topic into two major components: oligonucleotide therapeutics as a new anticancer drug for undruggable proteins and non-coding RNA. Most of the references cited in the manuscript are up-to-date. The manuscript is generally systematic, well written, and focused, except the citations are insufficient in the introduction and Challenges (part 4 and part 5). Overall, the impact of this review is profound and might interest other readers; however, there are still some minor faults to be improved.
1. We understand that miRNA is a prevalent study. However, considering the broad readership of this journal, we strongly recommend that please define acronyms and initials on their first use by giving the abbreviation in parentheses after the complete terminology and avoid acronyms in the abstract unless the abbreviation is used multiple times in the abstract. For example, in Figure 1, the abbreviations are well defined. Also, please do not use unnecessary abbreviations unless they appear at least three times in the text. Please check all the abbreviations you used in the manuscript, ex. pharmacokinetics/s. It is confusing for readers without following these rules.
2. It is a pity that the citation in the introduction and the challenges part is very poor, and references are missing in the context. Here are a few examples in the introduction part:
"Cancer incidence and...." in line 32.
"…because drug resistance and emerges and tumor evolves" in line 36.
"However, it is estimated that a large….as the small molecules" in line 45.
"…it has the double advantage…" in line 55.
"The ncRNAs are also …and linked to cancer." Rephrase it and also give it a reference, please.
The citation is the key to a relevant review article. Please fix it.
3. There is still some other typo. Please check.
4. Authors may consider renaming the tile since the precise information in state-of-the-art clinical studies is provided in the manuscript. It will attack more readers and even citations for the publication soon.
5. Please move Line 427 to Line 441 to the introduction. It is not appropriate to provide such a lengthy and fundamental statement in part 5 for discussing the challenges and solutions. Also, from Line 442 to Line 457, the citation is very poor again. I think this is one of the key elements of the whole review.
6. From Line 40 to Line 56, it is kind of bizarre that there is one sentence for one paragraph. Think of putting them all together to form a complete paragraph.
Reviewer 2 Report
The paper entitled “Precision anti-cancer medicines by oligonucleotide therapeutics targeting undruggable proteins and non-coding RNAs” is a review where the authors gather data regarding the use of oligonucleotides as new anti-cancer drugs by targeting RNA or DNA sequences of genes involved in cancer progression. They provide a list of some of the therapeutic oligonucleotides that are under clinical trial for tumor treatment. They focus on therapeutic oligonucleotides that target genes coding for proteins that are difficult to be targeted by conventional chemotherapeutic drugs (undruggable) or those that target ncRNAs relevant in cancer disease.
The contents of this paper resemble those recently published by Xiong and coworkers (Recent advances in oligonucleotide therapeutics in oncology. Int. J. Mol. Sci. (2021)), and some improvements should be addressed before publication. Here I provide some recommendations.
1. INTRODUCTION
- Many assertions that are displayed in the introduction are not supported by a reference. Each paragraph should be ratified by, at least, a reference that supports its contents.
- Line 52. The use of the term “sponge” in the introduction without a deeper explanation may be confusing.
- Line 67. “15 oligonucleotides therapeutics have been approved in USA for the treatment of different rare diseases, but none of them is used for cancer”… Obvious!!. I think that the authors mean that “at the moment there is no approved oligonucleotide-based cancer therapy in the USA”. What about their use in other countries?.
2. OLIGONUCLEOTIDE THERAPEUTICS
- Line 87. In my experience the level of specificity of the binding is not “that high”. Indeed, there is a high risk of off-targets.
- Figure 1. I guess that ASO stands for “Antisense oligonucleotide” but it is not explained in the text. It should be indicated.
On the other hand, if I am not mistaken ALL the therapeutical oligonucleotides are actually antisense oligonucleotides (they must bind the complementary RNA or DNA of their targets). I guess that the difference between ASO and miRNA or siRNA relies in the “natural” (known preexisting miRNAs or siRNAs) or “synthetic” (designed to target a region of interest) origin of the sequence.
It should be explained in the text.
- Table 1.
o It contains a summary of the “MOST RELEVANT” oligonucleotides in clinical trials, but there are not clear CRITERIA explaining their relevance.
o Which is the source of this information?. Clinicaltrials.gov?
o Are there conventional chemotherapeutic drugs available for some of the targets listed in the table?. It could be interesting to point it out.
o There are a lot of clinical trials in phase 1, but it is not clear how many of them are currently being evaluated and which have been definitely suspended or terminated and will not go beyond. It could be interesting to specify the current state of the clinical trial.
- It is not clear the reason why only MYC and RAS families are considered as the most relevant. According to table 1 several clinical trials have been performed targeting BCL2 but they are not described in the text. Only STAT3 as target of a therapeutic oligonucleotide has been included. What are the reasons that support such selection?
2.1 MYC GENE FAMILY
2.1.1 Targeting MYC.
There is a duplication in table 1 for the oligonucleotide DCR-MYC. It appears as being both ASO and siRNA. Only one should be listed in the table.
2.1.2 Targeting MYCN
- Line 125. Specify the acronym MNA as “MYCN amplification”.
- The paragraphs comprising lines 123 to 196 contain an extensive description of some interesting results obtained by the authors in previous studies regarding the use of the peptide nucleic acid BGA002 to block MYCN transcription. However,
- The extension of the text is disproportionated compared to other sections in the paper (too long).
- BGA002 is not under clinical trial yet and it is not listed in table 1. So, it does not meet the criteria considered in the abstract.
- Several paragraphs of this section are repetitive.
- As it is intended to be a review paper, any mention to own results should be avoided (e.g., “We also showed” in line 173, “We also found” in line 180, “Our study shows” in line 186) and be explained in a more general way.
I don’t mean that this section should be removed but the text should be rewritten to keep the uniformity and mention the results in a more discreet way.
2.2 RAS GENE FAMILY
2.2.1 KRAS
- The most recent reference provided sustaining the effectiveness of the oligonucleotide siGL2D-LODER is of 2016. Is the clinical trial active currently?
- There are several oligonucleotide strategies targeting KRAS under clinical trial listed in table 1 that have not been explained in the text. Is there a reason why they have been excluded?
- Line 221. The meaning of the acronym PLGA
2.3 STAT3
- Line 252. CpG-Stat3siRNA seems to be a new approach to regulate both STAT3 and TLR9 genes with a single drug. It is a relevant advance that should be explained deeper. Furthermore, nothing has been said about TLR9 and its relevance in cancer progression (at least not in this section…). In my opinion this paragraph could be improved.
3. NON-CODING RNAS
The paragraph introducing the characteristics of ncRNAs (lines 264 to 273) could also be improved. It should be clear that:
1) There are two classes of ncRNAs: structural (rRNAs, tRNAs, snRNAs,…) and regulatory.
2) The regulatory ncRNAs are classified as lncRNAs or sncRNAs (miRNAs, siRNAs and Piwi-interacting RNAs) according to their size.
3) Regulatory ncRNAs are the targets for oligonucleotide therapeutic strategies.
- Line273: “The expression of some RNAs is dysregulated and contributes to the development and progression of various diseases, including cancer”.
§ “RNAs” should be changed to “ncRNAs”, I guess…
§ Provide references to reviews where this topic is discussed.
- Line 284: “20 clinical trials have been initiated…”
§ Are all those 20 trials involved in cancer therapy? Are they listed in table 1?.
3.1 microRNA
- Line 293: Refer to table 1
- Four clinical trials have been terminated or withdrawn but the reasons are discussed only for two of them. Nothing to say about the others?.
3.1.1 Cobomarsen
- Line 307: “the expression of direct targets of miR-155 was downregulated”. How is it possible?, if cobomarsen is an inhibitor of miR-155 then their direct targets should be upregulated… Furthermore, in reference number 65 it is clearly mentioned that “cobomarsen….derepressed direct miR-155 target genes”.
3.1.3 MRX34
-Line 349. Authors mention that liposome formulation depends on amphoteric lipids. What is the relevance of this property (pH changes) in cancer therapy?.
4. CHALLENGES
- There is an absolute lack of bibliographic references in this section!.
- How many clinical trials currently rely in the use of modified oligonucleotides?.
Reviewer 3 Report
The topic of the manuscript is very interesting since the antisense oligonucleotide strategy is supposed to target proteins for which there is no possible immunotherapy available. The authors describe different strategies using oligonucleotides blocking translation from different genes which are important in human carcinogenesis.
I have doubts about the Fig 2. The figure is important in the review but it is almost identical (no information about copyright agreement) as in Quemener AM at al., Small Drugs, Huge Impact, Molecules 2022. This paper is not in the references list. It should be solved out whether it is suspicion of plagiarism.
The authors did not explain the differences among the three generations of antisense oligonucleotides in the sense of anticancer therapy. The readers should understand the reasons of synthesis of new generations of compounds.
Table 1 collects the information about clinical trials using oligonucleotides in cancer therapy. The clinical trials are mainly in Phase 1 or 2, few are in Phase 3. It is important information that this is new technology in cancer treatment and it is not in routine use yet. However, antisense therapy is already in clinical use for other diseases, the authors mention that.
There are some editing errors.
Round 2
Reviewer 1 Report
Thank you for contributing such a great work to provide very well organized review article.
Reviewer 2 Report
All my suggestions have been properly modified in the current version of the paper. Thanks for your kind explanations.
Reviewer 3 Report
I do hope that the identical formula of antisense nucleotides in the other paper was solved.
The authors described the differences among the three generations of antisense oligonucleotides designed for cancer treatment.